# Public Policy and Citizens' Attitudes towards Intelligent and Sustainable Transportation Solutions in the City—The Example of Lodz, Poland

Aldona Podgórniak-Krzykacz and Justyna Przywojska *

Department of Labour and Social Policy, University of Lodz, 90-136 Lodz, Poland
* Correspondence: justyna.przywojska@uni.lodz.pl; Tel.: +48-42-635-52-44

**Abstract:** This article focuses on residents' perception of smart and sustainable transport in the city, and on local and central government policy towards sustainable and smart mobility transition, using Lodz, a city in Poland, as an example. Its purpose is to examine residents' opinions about the development of a sustainable, intelligent transportation system (ITS) for the city and learn about their awareness of the benefits of ITS development. The article discusses findings of a survey conducted in 2022 on a sample of 250 residents of Lodz. The data obtained from the survey were subjected to statistical analyses: correlation, exploratory factor analysis, and linear regression. The results suggest there is a correlation between the assessment of a sustainable and intelligent transportation system and residents' perception of benefits that it offers. In addition, government, regional, and municipal transportation strategies were qualitatively analysed to identify the preferred trends featuring in the development of the transportation system and services in Lodz. The analysis confirmed fragmentation of programme-related actions that promote sustainable mobility, and the lack of participatory and educational measures targeting the users of transport systems. The results obtained can be used to delineate the directions of interventions in urban transport policy and sustainable development.

**Keywords:** sustainable smart mobility; urban transportation system; sustainable transportation solution; intelligent transport system; benefits of ITS; transport policy; Lodz

## 1. Introduction

The idea of sustainable development has been widely present in research works, as well as political and social programmes, particularly since the publication of the Brundtland Report where the concept was defined as "development that meets the needs of the present without compromising the ability of future generations to meet their own needs". It is generally believed that the concept of sustainable development rests on three pillars: economic, social, and environmental sustainability [1]. In principle, the transportation sector impacts sustainability across all these pillars (Table 1), which means that sustainable transport solutions need to be carefully planned and implemented. Consequences of the absence of sustainability in transport include environmental effects (e.g., GHG emissions), but also social (e.g., social exclusion and isolation) and economic (traffic congestion cost) consequences [2]. At the same time, it is assumed that sustainable transport derives directly from the idea of sustainable development, and can be described by factors leading to its unsustainability and by preventive and remedial solutions to negative externalities in transport [3,4].

**Table 1.** Impact of transport upon sustainability pillars.

| Economic | Social | Environmental |
|---|---|---|
| Traffic congestion | Inequity of impacts | |
| Mobility barriers | Mobility disadvantages | Air and water pollution |
| Accident damages | Human health impacts | Habitat loss |
| Facility costs | Community interaction | Hydrologic impacts |
| Consumer costs | Community liveability | DNRR |
| DNRR | Aesthetics | |

DNRR: Depletion of non-renewable resources. Source: [5]

Environmentally sustainable transportation systems consider the need to protect natural resources, strive to minimise pollution, and mitigate the impact of transport on ecosystems by, e.g., directing user attention to public transport, bicycling, and walking, which are generally assumed to be more energy efficient, and less polluting [6]. Economically sustainable transportation systems are affordable, efficient, allow users to choose the means of transport, and support the economy [7]. Transportation systems that are socially sustainable are not only efficient but crucial for the reduction of poverty and social exclusion, by ensuring access to markets, employment, education, and basic services [8]. Seen within the broader context, a sustainable and integrated transportation system encompasses the above dimensions and plays an important role in achieving sustainable development goals.

To accomplish these goals, local authorities need to adopt a sustainable, integrated, and innovative transport policy that optimises the use of different modes of passenger and cargo transport. The challenge consists of meeting the needs of citizens by offering them accessible, reliable, efficient, and safe transport. A sustainable transportation system in a city requires focusing on the needs of passengers and on shaping sustainable transport behaviour patterns through public policy [9]. This means offering solutions that are cheap, accessible, healthy, environmentally friendly, reliable, and safe. Automated and network-based multimodal mobility is also expected to play an increasingly important role, alongside intelligent traffic management systems implemented in cities within the digitalisation framework. The right mix of measures should aim to address urban congestion and improve public transport services [10].

In particular, sustainable mobility requires the following [11–13]: (a) optimal use of technology, including investment in technologies related to the means of transport, IT systems, and the transport system itself; (b) price control and pricing that reflects the external costs of transport in actual cost of travel through higher fuel prices or some form of road-user charges; (c) land-use development, including planning and regulations which should be integrated so that the physical constraints and development patterns could support shorter-distance travel; (d) clearly targeted personal information, including awareness raising, persuasion, and personal marketing. Approval is an essential (although often neglected) component of sustainable mobility. On the other hand, an ideal vision of "smart transition" presents mobility of the future as a personalised "service" available "on demand", where individuals have immediate access to a seamless system of clean, green, efficient, and flexible transport meeting all their needs [14].

Thus, a sustainable and smart urban transportation system encompasses innovative and sustainable transportation solutions [15], which often (but not always) deploy ICT technologies and can be grouped into several categories [16–18]. The first group brings together modern solutions in public transport that include the introduction of environmentally friendly fleet vehicles and fuels (e.g., electric vehicles) or automated driving vehicles, as well as interventions which improve the quality and integration of public transport services (e.g., the introduction of an integrated ticketing system, e-ticketing in public transport, or integration of public transport with other mobility options). The second group is made up of modern solutions for private transport. It includes the use of environmentally friendly vehicles by private users (e.g., electric cars) and putting in place transport solutions which change user transport behaviour to a more sustainable and multimodal one (e.g., vehicle

sharing: public transport, bike sharing, scooter sharing, car sharing; on-demand services: ride hailing, carpooling, etc.; Mobility-as-a-Service; or eco-driving). The third group includes infrastructure and policies that support sustainable and smart mobility (e.g., park and ride systems, intelligent parking systems, bike lanes, bus lanes, pedestrian zones, economic instruments and taxes for reduction of emissions from transport, or regulated access to various zones in the city's low emission zone). The next area of smart transportation covers intelligent transport systems (ITS) using intelligent sensing, communication and computing techniques for automatic urban traffic and travel information management, passenger information management in public transport, and vehicle control [17].

It is believed that the implementation of the above sustainable and smart transport solutions ensures better safety to users, reduces air and noise pollution, diminishes congestion, improves the efficiency of the use of infrastructure and vehicles, increases the speed with which we travel, and guarantees a better match between the services and user needs or provides a wider range of mobility services [14,16,18]. However, an analysis of the literature allows us to conclude that the effects of the implementation of smart and sustainable transport solutions on urban sustainability are not widely published, and are usually based on simulation and model studies rather than on field studies of actually implemented solutions [19]. Existing analyses confirm positive environmental impact of smart transportation, as it effectively mitigates the growth of $CO_2$ emissions [20]. Environmental and socio-economic benefits also feature in studies on ITS [21]. Evidence is also available for the environmental and social benefits of shared mobility practised by city dwellers. For instance, Chen and Deng [22] demonstrated that car sharing reduces trips in private vehicles in the city, and can reduce vehicle emissions as well as the occurrence of accidents and traffic congestion in the urban network to some extent. It can also improve convenience and promote city sustainability through the improvement of urban conditions in terms of social and environmental aspects. Yet, researchers' findings are not unambiguous when it comes to benefits resulting from electric mobility on sustainable development. The use of Life Cycle Sustainability Assessment in these analyses confirms the potentially positive impact of electric vehicles on the environment (reduction in global warming potential, air pollution, and photochemical oxidant formation) [23]. On the other hand, the same studies have shown that at the social level (in terms of employment generation, compensation of employees, and taxes) performance indicators were better for internal combustion vehicles than for electric ones, and that adoption of electric vehicle alternatives does not lead to better macro-economic indicators due to the differences in life cycle cost between these options. In addition, researchers highlight the problem of social exclusion produced by the designation of Low Emission Zones [24], or the reproduction of transport exclusion and transport poverty caused by the digitization of transport services [25,26].

Subject matter literature emphasises that consumers are a key component of the smart mobility concept [15,27]. It means that transport solutions should consider their needs, which, in turn, should ensure benefits to people from smart mobility solutions. Appropriate mobility behaviour of the urban population could help in achieving some of the above benefits [28]. Thus, one could conclude that demand for smart mobility solutions depends on their availability in cities and on how they are perceived by the users. For instance, surveys by Ahmed et al. [29] confirmed a significant relationship between consumer attitudes and an intention to adopt smart mobility solutions. The more positive consumers' attitudes to transport technology are, the readier they are to embrace a particular service. The literature also mentions some other socio-economic variables important to the use of smart mobility solutions, such as gender [30,31] or age [14,32,33].

The role of social participation is also highlighted as essential if sustainable and smart transport-related projects are to win political support and change residents' behaviour, which are fundamental for their success. In particular, community participation through strong NGOs may help in changing urban systems in a more sustainable way [34,35]. Advocates of cooperation-based planning point to the significant potential of community engagement in drafting mobility plans. The aim is to foster social capital, improve social

coherence, win user trust and approval, achieve better environmental outcomes, and improve the organisation of transport services [36,37]. Sustainable Urban Mobility Planning (SUMP), an idea developed to support local authorities in building new urban mobility strategies, assumes broad participation of city residents [38]. SUMP relies on strategic plans aimed to address the mobility needs of inhabitants and businesses in and around cities to ensure better quality of life. This new planning concept allows solving transport-related issues in a more sustainable and comprehensive way [39,40]. This approach, promoted by the European Commission, sets out, among other things, the key principle according to which society should be involved in transport-related planning from the very beginning of the process, and not only when plans are almost completed and only minor amendments are feasible through public consultations. However, urban mobility transition requires not only the engagement of residents in planning the change, but also making them a part of it by changing residents' lifestyles and mobility habits. Changing human behaviour usually occurs through raising awareness of their choices, behaviour patterns, and their consequences [41–43]. Measuring human activity related to the use of resources and providing information to transport system users to motivate them to carry out change are helpful here. Education on desired behaviour patterns and available transport solutions is also important [44].

The aim of this paper is to compare the determinants of smart urban mobility transition representing the political and social environment. At the political level, the aim is to identify solutions and instruments proposed by national, regional, and local strategies to foster the development of smart and sustainable urban mobility. At the social level, the aim is to identify factors determining the use of smart mobility solutions. The combination of these two perspectives allows for a comprehensive analysis of the determinants of smart mobility transition and is new in the literature. To achieve our goal, we used two research methods: content analysis of national, regional, and local public strategies and a survey of users of the transport system in Lodz.

## 2. Materials and Methods

### 2.1. Research Assumptions and Methods

Based on the literature review carried out in the previous section, we assume that smart transition of the urban mobility system requires the development of an intelligent, sustainable transportation system that is attractive to residents and stakeholders who want to use it or are "forced" to use it. Therefore, our study is focused on two aspects: public strategies for smart mobility transition, which identify the conditions, incentives, priorities, incentives and tools of coercion, and the opinions of users of smart transport solutions on their development and perceived benefits.

Our study concerns Lodz, one of the largest Polish cities, and attempts to provide answers to the following research questions:

1. What solutions, if any, leading to a sustainable smart mobility transition have been proposed in national, regional, and local transport development strategies?
2. What are the opinions of Lodz inhabitants about the development of a sustainable intelligent transport system in the city?
3. Do the inhabitants perceive benefits from the implementation of intelligent transport systems?
4. Do the opinions of Lodz residents on sustainable and an intelligent urban transport system depend on their perception of the benefits offered by ITS?

In the first step of this study, we described the features of the research area, i.e., the transportation system in Lodz, and reviewed sustainable and smart solutions implemented within this system (Section 3). We analysed research papers, data and publications from the Roads and Transport Authority of the City of Lodz, as well as the market for e-mobility services in the city.

In the second step of this study, we carried out content analysis of strategies binding at each (national, regional, and local) level of public governance in Poland and their

assumptions for government and local government transport policy for cities. The following documents were selected for this study:

- Strategy for Sustainable Transport Development until 2030;
- National Urban Policy 2030;
- Lodz Voivodeship Development Strategy 2030;
- City of Lodz Development Strategy 2030+;
- Sustainable Development Plan for Public Transport in Lodz until 2025.

The main objective of the content analysis of the above-mentioned documents was to identify the key principles of urban transport policy and actions designed for the transition towards sustainable and intelligent urban transport systems. The strategies were analysed using the tool developed by the authors on the basis of the literature review (Introduction Section). Our research assumptions rely on the findings of other authors [11,45,46], who proposed the following dimensions of transition towards sustainable mobility: (1) policies aimed at improving public infrastructure and provision; (2) policies facilitating cycling and walking; (3) policies regulating the use of private cars; (4) policies aimed at changing attitudes and behaviours; (5) spatial planning strategies; (6) investments in roads and railways; (7) policies involving the optimal use of technology, including investment in technology for means of transport, information systems and the transport system itself; (8) participatory mobility planning. Based on these we selected the following detailed objectives for the analysis of the content of strategies: (1) identification of the principles underpinning the transport policy in the context of transition to sustainable development, and (2) identification of actions shaping sustainable and intelligent transport systems in cities in the following categories: transport-related solutions, planning instruments, and instruments shaping residents' transport decisions. Results are presented in Section 4.1.

In order to learn what Lodz residents think about the sustainable, intelligent transport system developed for the city and the perceived benefits of developing ITS, we conducted a quantitative survey using an online survey questionnaire. The survey was conducted between December 2021 and January 2022, and it covered 250 users of the Lodz transport system, which allows for the generalization with the margin of error of approx. 6%. The characteristics of the research sample are included in Table 2. Results of quantitative studies are included in Section 4.2 which discusses the distribution of respondents' answers. Basic descriptive statistics were also deployed. The strength of a relationship between data was measured with Spearman's rho correlationcoefficient or Pearson's linear correlation coefficient. An exploratory factor analysis was conducted using the principal components analysis as a method to extract common variability, and Equamax orthogonal rotation and the Kaiser criterion were deployed to determine the number of factors. In addition, logistic regression was used to identify factors important for the assessment of a sustainable and intelligent transportation system.

**Table 2.** Sample structure.

| Category | | n | % |
|---|---|---|---|
| Total | | 250 | 100.0 |
| Gender | Woman | 132 | 52.8 |
| | Man | 118 | 47.2 |
| Age | under 20 years of age | 21 | 8.4 |
| | 21–30 | 180 | 72.0 |
| | 31–40 | 20 | 8.0 |
| | 41–50 | 20 | 8.0 |
| | over 50 years of age | 9 | 3.6 |
| Labour force participation | Active | 94 | 37.6 |
| | Inactive | 150 | 60.0 |
| | Decline to answer | 6 | 2.4 |

**Table 2.** *Cont.*

| Category | | n | % |
|---|---|---|---|
| Monthly disposable income | Not more than PLN 1000 | 56 | 29.6 |
| | 1001–2000 | 42 | 22.2 |
| | 2001–3000 | 39 | 20.6 |
| | 3001–5000 | 33 | 17.5 |
| | More than PLN 5000 | 19 | 10.1 |
| | Decline to answer | 61 | 24.4 |
| Car ownership | Yes | 157 | 62.8 |
| | No | 93 | 37.2 |

Source: own elaboration.

*2.2. Measurements*

The sustainable, intelligent transport system in the city was assessed against 19 items (Table 3) selected based on the literature review [16–18]. Respondents replied using a 5-point Likert scale.

**Table 3.** Measurement scale for the development of sustainable, intelligent urban transport system.

| | Items | Scale |
|---|---|---|
| **Y1** | Assess the development of intelligent transport systems | |
| **Y2** | Assess the development of modern transport services | |
| **Y3** | Assess the development of mobile apps for transport | |
| **Y4** | Assess pedestrian lanes | |
| **Y5** | Assess woonerfs | |
| **Y6** | Assess bike lanes | |
| **Y7** | Assess public transport stops | |
| **Y8** | Assess tram tracks | |
| **Y9** | Assess the development of transport interchanges | |
| **Y10** | Assess road infrastructure | Very poor, poor, moderate, good, very good, I have no opinion |
| **Y11** | Assess ring roads | |
| **Y12** | Assess motorways and expressways around the city | |
| **Y13** | Assess car parks in the city | |
| **Y14** | Assess car parks at the outskirts of the city | |
| **Y15** | Assess EV charging infrastructure | |
| **Y16** | Assess the clarity and availability of timetables | |
| **Y17** | Assess the clarity and availability of bus/tram route maps | |
| **Y18** | Assess the clarity and availability of digital displays at stops | |
| **Y19** | Assess possibilities to interchange between public transport and vehicles rented by the minute | |

Source: own elaboration.

An exploratory factor analysis was used to identify a set of unobserved factors (F1–F6) and to test the validity and comprehensiveness of this measurement. Cronbach's alpha coefficient was calculated to assess the reliability of the measurement. In the next step, a summary score was calculated by summing up all respondents' scores.

Respondents' perceptions of the benefits of intelligent transportation systems were measured against 5 items selected on the basis of the literature review [14,16,18] (Table 4). In their answers, respondents used a 7-point Likert scale. Respondents were provided with the definition of intelligent transportation systems, explaining that they are a combination of a variety of advanced technologies (in the field of telecommunication, IT, automation, and measurement) as well as transport management and service technologies applied with a view to ensure innovative and user-friendly transport services [47].

**Table 4.** Measurement scale for the perception of benefits of intelligent transport systems.

| | Items | Scale |
|---|---|---|
| **X1** | Impact of intelligent transport systems upon the shortening of travel time | |
| **X2** | Impact of intelligent transport systems upon the improvement of road safety | Definitely not, not, rather not, |
| **X3** | Impact of intelligent transport systems upon the reduction of traffic congestion | rather yes, yes, definitely yes, |
| **X4** | Impact of intelligent transport systems upon the improvement of travellers' comfort | I have no opinion |
| **X5** | Impact of intelligent transport systems upon the reduction of environmental pollution | |

Source: own elaboration.

Again, a summary score of perceived benefits was calculated by summing up the scores given by the respondents. For both summary variables (assessment of the development and perceived benefits), a correlation coefficient was calculated and a linear regression model was estimated. The following control variables were included in the model: gender, age, labour force participation, monthly disposable income, and car ownership.

## 3. Characteristics of the Research Area

Lodz is the fourth city in Poland by population (690.1 k inhabitants) and by area occupied (293.3 km$^2$). As a regional capital, the city performs metropolitan functions and hosts many public institutions, meaning it faces the challenge of ensuring access to public transport within the city, agglomeration, region, and the country. Lodz is situated in the centre of Poland and it enjoys good transport connections via the A1 and A2 motorways and the S8 and S14 expressways, which act as its ring road. The transport network in Lodz comprises roads, railways, passenger and freight transport nodes, and the airport. The road and street system of the city is composed of the following elements: a section of the A1 motorway (in the eastern part of the city), four national roads, and two regional (voivodeship) roads. Lower category roads form a rectangular system stretching between the national and regional roads. The transport system in Lodz assumes a grid-like shape in the city centre and the road layout becomes less regular the further we move from the inner city [48]. Tramway tracks also belong to the road system; their total length is approximately 142 km. Tram routes are particularly dense in the inner city. Another characteristic feature of the Lodz tram system is the positioning of a high proportion of the rails in the middle of narrow streets in the city centre.

Local public transport, operated by the municipal operator MPK-Łódź Spółka z o.o. (MPK), consists of two complementary bus and tram systems, which on many routes (especially in the inner city and along major roads) are interchangeable [49]. Transport services are rendered by 80 bus routes (including 7 night routes) and 22 tram lines. MPK has a fleet of 459 tram cars and 435 buses. The local public transport subsystem serves the city of Lodz and the neighbouring municipalities. Every day, MPK vehicles transport approximately 740,000 passengers [50]. Railway transport, whose services are offered by both the regional carrier and the Polish Railway PKP IC company, responsible for supra-regional accelerated connections, is of great importance for passenger transport in the Lodz agglomeration area. The regional transport service provider is Łódzka Kolej Aglomeracyjna sp. z o.o. (Lodz Agglomeration Railway company), established on 10 May 2010. It runs transportation services based on a contract established with the Local Government of the Lodzkie Voivodeship, which holds 100% of the company shares. According to researchers, the Lodz Agglomeration Railway mainly caters to the transport needs of the towns neighbouring Lodz, which is due to the fact that the railway tracks run mainly on the outskirts of Lodz, outside the most populated area [51,52]. The analyses of the possibilities of using the Lodz Agglomeration Railway as a component of the urban public transport also pointed to the poor interconnection of the stops [53].

At present, investment projects are carried out within the city that aim to expand the railway infrastructure. The objective is to encourage more use of the railway for transport within the agglomeration and inner-city by upgrading railway tracks, stations, and stops, and by building new stops as well as modernizing the rolling stock. The largest investment

in rail transport in Lodz is the construction of an underground Fabryczna station and a cross-town tunnel to run through the city centre. Local, regional, and long-distance trains are to pass through the cross-town tunnel. The project envisages the construction of three stops in Lodz. This will enable an underground connection between stations located at different ends of the city and facilitate getting around Lodz.

The city is one of the most congested in Poland and in Europe, due to the high mobility of its inhabitants and the daily influx of people from the neighbouring municipalities. On top of that, its over-congested road and street network must cope with the ongoing renovations of tramway tracks and road works, as well as the construction projects carried out as part of the revitalisation process which limit the network's capacity. Another transport problem in the city is the lack of a sufficient number of car parks, also in the park and ride scheme. The city has put in place a number of sustainable solutions to improve the efficiency of the transport system. Some investment projects are intended to develop safe pedestrian routes, woonerfs (currently 18), and bike routes (238 km of cycling routes equipped with cycle counters); there are also infrastructure investments dedicated to public transport, including the renovation of railway tracks (11 km), installation of multimedia bus shelters, purchase of electric rolling stock (17 buses and 7 charging stations), separation of bus lanes also dedicated to low-emission cars (11 km), and investments for electric car users (28 chargers, parking spaces) [54]. One of the infrastructural solutions supporting the propagation of public transport is the construction of intermodal transfer hubs. The "Łódź Fabryczna" and the "Łódź Kaliska" railway stations meet the requirements of a mobility hub. An important interchange, but mainly for public transport users, is the Tramway Station "Centre" on the W-Z (East–West) route.

One of the smart solutions implemented in Lodz public transport is the Dynamic Bus Information System, which operates at more than 140 stops. The boards are used for real time display of times at which trams or buses arrive at a particular stop, as well as current breakdowns and changes to the service. Real-time tracking of public transport vehicles is also possible through mobile applications. Tickets can be purchased at ticket vending machines installed at stops and in vehicles (including e-ticketing) using the Open Payment System as well as via the mobile app. Urban public transport tickets are integrated with tickets of the Lodz Agglomeration Railway.

In Lodz, the ITS system was introduced back in 2014. It includes the Smart Coordinated Adaptive Traffic System (SCATS), operating at 234 intersections [55]. It enables traffic lights to be controlled and optimised. Another subsystem is the Mobile Information System for Drivers consisting of nine variable message boards. The boards display information about current traffic jams, accidents, breakdowns, and planned or ongoing road works that may disrupt traffic. The ITS also includes a video surveillance system that automatically recognises number plates, as well as the Mobile Information subsystem. Another subsystem is the tunnel control system.

Lodz also offers certain modern shared transport services, such as bike sharing, car sharing, scooter sharing, and moped scooter sharing. Between 2021 and 2024 the bicycle system is being operated by Homeport company, which provides 1510 bicycles at 151 stations. There are four companies in the city offering electric scooter rental: Volt Scooters, Blinkee.city, Bolt, and Lime, and one company offering scooter rental (Blinkee.city). The carsharing service was introduced in the city in 2018 by Easyshare. Today, there are two carsharing companies operating in Lodz: Traficar (200 vehicles) and Panek (200 vehicles). The idea of carpooling is also known in Lodz; these services are offered by global providers such as BlaBlaCar. For the slightly different formula of ride hailing, the inhabitants of Lodz can use the services of, e.g., Uber or Bolt. In addition, residents have the opportunity to use applications that integrate shared mobility measures within the VooM application.

The city actively promotes sustainable urban transport in order to change the transport behaviour patterns of residents. Among others, Lodz has joined the European action promoting sustainable and green transport. In 2021, its slogan was "Enjoy sustainable mobility. Take care of your health." Events were prepared in the city to familiarise residents

with the history and functioning of public transport, and information was provided on alternative modes of transport available in the city and their possible use [56].

## 4. Results

*4.1. Shaping Sustainable and Intelligent Transport Systems in Cities in the Light of Government and Local Authority Strategies*

The provisions of official public strategic documents reflect the assumptions of transport policies adopted by the government and local authorities. The analysis of the content of government and local government strategies was used to identify the key principles behind the transport policy and priority actions for sustainable urban transport development. Its results are presented in Table 5.

**Table 5.** Results of the analysis of government and local authorities' strategies in the light of transport policy and actions aimed at developing sustainable urban transport.

| Strategy | | Strategy for Sustainable Transport Development until 2030 | National Urban Policy 2030 | Lodz Voivodeship Development Strategy 2030 | City of Lodz Development Strategy 2030+ | Sustainable Development Plan for Public Transport in Lodz until 2025 |
|---|---|---|---|---|---|---|
| Principles | Mainstreaming the sustainable development principle | X | X | X | X | X |
| | Citizens participation in urban transport system planning and management | - | - | - | - | - |
| | Transport digitization | X | X | X | X | X |
| | Planned measures integrated by linking transport with socio-economic aspects, environment, space, management, and safety | X | X | - | X | - |
| | Spatial planning that considers universal design and reduces space occupied by transport | X | X | X | X | - |
| Actions shaping sustainable and intelligent transport systems in cities | Transport-related solutions | Integrating public transport system into functional areas and agglomerations | X | X | X | X | X |
| | | Improved transport infrastructure — X | X | X | X |
| | | Building a system for charging and fuelling low-emission vehicles | X | - | X | - | - |
| | | Fine-tuning the quality of public transport services | X | - | X | X | X |
| | | Shared mobility and green mobility in cities and in functional areas | X | X | X | X | - |
| | | Developing green transport lanes | X | - | - | X | - |
| | | Modern fleet | X | X | X | X | X |
| | | ITS | X | X | X | X | X |
| Planning instruments | SUMP | X | X | - | - | - |
| Instruments shaping residents' transport decisions | Price and fiscal mechanisms | X | - | - | - | - |
| | Restrictions/bans on access for passenger cars | X | X | - | X | - |
| | Educating citizens | - | - | - | - | - |
| | Promoting sustainable, smart transport options | X | X | - | - | X |

X means that the item analysed is included in the strategy. - means that the item analysed is not contained in the strategy. Source: own elaboration.

The analysis of the content of government and self-government strategies leads to the conclusion that their key values make reference to sustainable and smart territorial development. The strategies identify the main threats and challenges to the development of Polish municipalities and declare that their mission is to create conditions for improving the quality of life of the inhabitants, ensure inclusion, and build resilience to observed climate change. However, the directions of actions proposed in response to the diagnosed challenges in the area of transport do not take into account some of the proposed policy components for the development of sustainable and intelligent transport systems. Identified shortcomings are as follows:

1.　Failure to recognise the potential for using instruments of a pricing and fiscal nature in local and regional policies. The strategies analysed at the regional and local level do not contemplate the possibility of using pricing mechanisms to develop and shape residents' sustainable transport behaviour patterns. Only the government Strategy for Sustainable Transport Development until 2030 proposes solutions in this regard, indicating pricing and fiscal mechanisms. Changes in the tax system are proposed to ensure reward for the purchase, ownership, and use of vehicles exerting lower environmental pressure (both when it comes to emissions and consumption of energy carriers). Within the framework of pricing instruments, the formulation of a modern and pro-environmental parking policy was proposed that would reward the purchase,

ownership, and use of vehicles that do not overburden the environment. On the other hand, solutions put forward in the National Urban Policy 2030 are of an indirect nature, which means that they are not aimed directly at changing user transport behaviour in response to financial incentives. The document provides only suggestions concerning the use of financial mechanisms promoting the implementation of desired solutions (launching funds for these units), e.g., the development of SUMPs or the expansion of cycling routes by the local governments.

2.　Omission of the user perspective in sustainable mobility planning. None of the documents analysed proposed measures that would engage residents in the planning and management of the transport system. Only government strategies (Strategy for Sustainable Transport Development 2030 and National Urban Policy 2030) indicate the need for SUMPs being developed by local authorities, but these programmes are optional in Poland. The planning approach based on SUMPs assumes the widest possible participation of residents in planning processes. Unfortunately, the authorities of Lodz have not drafted a SUMP, nor did they propose activities in this area in their development strategy. Additionally, the regional development strategy does not foresee measures that would support municipal authorities in implementing this planning approach or prompt them to develop SUMPs.

3.　Lack of activities in the field of education. The analysed strategies basically neglect information policy, education of residents, or persuasion. None of the analysed strategies pay attention to issues such as residents' knowledge of sustainable mobility and intelligent transport solutions or the formation of skills, attitudes, and, consequently, competences in sustainable and smart mobility. Only the Sustainable Development Plan for Public Transport in Lodz draws attention to the need to develop appropriate marketing policy that would highlight environmental benefits the residents can experience if more use of public transport is encouraged. The promotion of sustainable, smart transport options was suggested in general terms in the government strategies.

4.　Overlooking green transport infrastructure. Actions promoting the development of green infrastructure accompanying transport routes can be found only in the government Strategy for Sustainable Transport Development until 2030. Some references to this issue are also made in the Strategy of Development of the City of Lodz 2030+; however, this is at a very limited scale—it is only planned for the development of infrastructure for pedestrian and cycling traffic.

5.　Lack of actions aimed at creating a system for charging and fuelling low-emission vehicles in the National Urban Policy 2030, in the City of Lodz Development Strategy 2030+, and in the Sustainable Development Plan for Public Transport in Lodz. This is particularly surprising, as the need to develop eco-mobility in cities and functional areas has been clearly formulated in all documents analysed. This will never be achieved without investing in the accompanying infrastructure that enables the charging and refuelling of green vehicles.

In addition, the results of the analyses of the government and municipal strategic documents suggest that the national strategies are comprehensive and create the conditions for an effective mobility transition towards sustainability and digitalisation. At the same time, their weakness lies in the lack of consideration of the principle of participation in building urban mobility systems. The National Urban Policy 2030 does not clearly distinguish areas of action concerning the improvement of transport infrastructure and the quality of transport services. Actions in this area are scattered and described in a fragmentary way, and as a side topic of other priority areas for the development of sustainable transport. This approach does not guarantee the integration of measures. Regional and local strategies, on the other hand, focus on hard measures, mainly investment projects, neglecting the impact of educational tools or information and communication technologies on transport behaviour. It is worth emphasising the specificity of the Sustainable Development Plan for Public Transport in Lodz, which refers exclusively to the plans for the development of the public transport sub-system within the city, taking into account agglomeration connections.

It adopts the perspective of the organiser and operator of public transport services and focuses primarily on accessibility, integration, and service quality. As a consequence, many aspects of urban transport system development have been neglected. On top of that, as in government documents, they omit the involvement of residents in the planning and implementation of transport-related solutions. It seems that the perspective of the customer—the user of the city transport system—has not been sufficiently taken into account in the strategies analysed. Apart from educational activities and promoting public participation, it is worthwhile to include systematic opinion surveys conducted among transport system users on the quality of the services offered and the shape of the infrastructure, and research into the transport behaviour of residents, including their attitudes, preferences, motives, needs, and readiness to accept technological solutions. The need for a systematic opinion survey of public transport passengers to diagnose the quality of services (without highlighting the issue of their digitisation) features only in the Sustainable Development Plan for Public Transport in Lodz.

### 4.2. Results of the Survey among the Citizens of Lodz

4.2.1. Assessment of the Development of an Intelligent Urban Transport System

Respondents assessed the development of Lodz's sustainable and intelligent transport system in relation to 19 aspects on a scale from 1 to 5, where 1 means very poorly developed and 5 means very well developed (Table 6). The evaluation of intelligent transport systems and modern transport services is high, with an average score of 4.28–4.60 (with max = 5), and the median reaching 5 (with a variation of 1.4–1.5). For these aspects, the percentage of high ratings—min. of 4—reaches 58–66%, with one in four or five respondents choosing the highest score. Timetables, bus/tram route plans, digital passenger information boards at bus stops, and ring roads are also rated highly, the average varying between 3.61 and 4.05, with the percentage of the highest scores ranging from several percent to as high as 27% (digital information boards at bus stops), with 2/3–3/4 of respondents selecting scores 4 or 5. Most of the analysed aspects are rated at an average level (median equal to 3, and mean mostly above 3). The situation is different with regard to two aspects: roads and car parks in the city, which half of respondents rated no higher than 2, with the average reaching 2.16–2.31.

**Table 6.** Assessment of the development of sustainable and intelligent transport system in Lodz.

| Var. | Item | Percentage of Responses | | | | | | Statistics | | |
|------|------|------|------|------|------|------|------|------|------|------|
| | | 1 | 2 | 3 | 4 | 5 | n.d. | M | Me | SD |
| Y1 | Intelligent transport systems | 13.2 | 25.2 | 3.2 | 38.0 | 20.4 | 0.0 | 4.28 | 5.00 | 1.53 |
| Y2 | Modern transport services | 8.0 | 18.8 | 6.4 | 41.2 | 25.6 | 0.0 | 4.60 | 5.00 | 1.43 |
| Y3 | Mobile apps for transport | 3.2 | 8.0 | 57.2 | 26.8 | 4.8 | 0.0 | 3.22 | 3.00 | 0.79 |
| Y4 | Pedestrian routes | 6.8 | 7.6 | 61.2 | 20.0 | 4.4 | 0.0 | 3.08 | 3.00 | 0.85 |
| Y5 | Woonerfs | 4.0 | 10.0 | 59.2 | 19.6 | 7.2 | 0.0 | 3.16 | 3.00 | 0.85 |
| Y6 | Cycling routes | 4.0 | 8.8 | 49.6 | 29.6 | 8.0 | 0.0 | 3.29 | 3.00 | 0.89 |
| Y7 | Public transport stops | 2.8 | 8.8 | 43.6 | 37.6 | 7.2 | 0.0 | 3.38 | 3.00 | 0.85 |
| Y8 | Tram tracks | 10.0 | 18.0 | 46.8 | 20.8 | 4.4 | 0.0 | 2.92 | 3.00 | 0.98 |
| Y9 | Transport interchanges | 4.0 | 10.0 | 46.8 | 30.4 | 8.8 | 0.0 | 3.30 | 3.00 | 0.91 |
| Y10 | Roads in the city | 32.8 | 29.2 | 28.0 | 8.8 | 1.2 | 0.0 | 2.16 | 2.00 | 1.02 |
| Y11 | Ring roads | 5.2 | 11.6 | 40.8 | 36.8 | 5.6 | 0.0 | 3.26 | 3.00 | 0.92 |
| Y12 | Motorways and expressways around the city | 2.4 | 4.4 | 32.8 | 46.4 | 14.0 | 0.0 | 3.65 | 4.00 | 0.86 |
| Y13 | Car parks in the city | 22.8 | 33.2 | 36.0 | 6.4 | 1.6 | 0.0 | 2.31 | 2.00 | 0.95 |
| Y14 | Car parks at city outskirts | 8.4 | 17.2 | 60.4 | 12.4 | 1.6 | 0.0 | 2.82 | 3.00 | 0.82 |
| Y15 | EV charging infrastructure | 18.8 | 16.8 | 55.6 | 6.4 | 2.4 | 0.0 | 2.57 | 3.00 | 0.95 |
| Y16 | Timetables | 2.8 | 4.8 | 29.6 | 42.4 | 17.2 | 3.2 | 3.69 | 4.00 | 0.92 |
| Y17 | Bus/tramway route maps | 2.0 | 6.0 | 32.8 | 42.0 | 13.2 | 4.0 | 3.61 | 4.00 | 0.88 |
| Y18 | Digital passenger information display boards at stops | 0.4 | 2.4 | 16.4 | 49.2 | 27.2 | 4.4 | 4.05 | 4.00 | 0.77 |
| Y19 | Possibilities to change between means of public transport and vehicles hired by the minute | 2.4 | 6.4 | 67.2 | 18.0 | 6.0 | 0.0 | 3.19 | 3.00 | 0.74 |

M—mean, Me—median, SD—standard deviation. Source: own elaboration.

The relation between variables describing the assessment of the development of a sustainable and intelligent transport system are mostly significantly correlated (Table S1).

Given the set of 19 aforementioned variables, an exploratory factor analysis was performed to assess the metric properties of the transport system development measurement tool. The analysis was conducted using the principal components analysis as a method to extract common variability, Equamax orthogonal rotation, and the Kaiser criterion to determine the number of factors. Both KMO = 0.808 and the results of the Bartlett's sphericity test ($p < 0.001$) confirm that the adopted set of variables is adequate for conducting a factor analysis. The 19 statements are grouped into six subscales (Table 7). The first of these (F1—intelligent transport systems and services) includes two top-rated aspects and explains about 28% of the variance in the latent variable. The second factor (F2—public transport passenger information system) explains nearly 9% of the variance in the latent variable and includes three issues: bus/tram route plans, timetables, and digital display boards at stops. The third factor (F3—national roads in the surroundings of the agglomeration) includes two elements: ring roads and motorways/expressways around the city, and explains 8.4% of the variance in the latent variable. It is worth noting the very high values of factor loadings for these items—between 0.733 and 0.847. The reliability of the scales obtained is also high; the Cronbach's alpha coefficient is 0.815/0.727 and 0.801, respectively.

**Table 7.** Results of exploratory factor analysis.

| | F1 | F2 | F3 | F4 | F5 | F6 |
|---|---|---|---|---|---|---|
| Intelligent transport systems | **0.847** | 0.107 | 0.079 | 0.209 | 0.156 | 0.080 |
| Modern transport services | **0.807** | 0.175 | 0.160 | 0.098 | 0.113 | 0.067 |
| Bus/tram route plans | 0.063 | **0.818** | −0.061 | 0.037 | 0.078 | 0.044 |
| Timetables | 0.072 | **0.782** | 0.092 | 0.131 | 0.064 | 0.175 |
| Digital information display boards at stops | 0.166 | **0.733** | 0.206 | 0.040 | −0.083 | 0.085 |
| Ring roads | 0.059 | 0.081 | **0.796** | 0.052 | 0.234 | 0.189 |
| Motorways and expressways around the city | 0.104 | 0.171 | **0.785** | 0.145 | 0.164 | 0.168 |
| Cycling routes | 0.143 | 0.001 | 0.126 | **0.784** | 0.017 | 0.193 |
| Pedestrian routes | 0.174 | 0.201 | 0.092 | **0.654** | 0.111 | 0.111 |
| Woonerfs | 0.332 | −0.062 | 0.345 | **0.556** | −0.250 | −0.030 |
| Car parks in the city | 0.048 | 0.138 | 0.050 | 0.326 | **0.700** | −0.229 |
| Roads in the city | 0.325 | −0.060 | −0.035 | −0.054 | **0.656** | 0.323 |
| EV charging infrastructure | 0.111 | −0.022 | 0.233 | −0.126 | **0.554** | 0.136 |
| Car parks at city outskirts | −0.063 | 0.075 | 0.229 | 0.475 | **0.530** | 0.012 |
| Changing between means of public transport and vehicles rented by the minute | −0.153 | 0.136 | 0.284 | −0.001 | −0.016 | **0.652** |
| Public transport stops | 0.236 | 0.170 | 0.094 | 0.360 | 0.150 | **0.616** |
| Tram tracks | 0.327 | 0.120 | −0.138 | 0.248 | 0.420 | **0.549** |
| Mobile apps for transport | 0.441 | 0.164 | 0.189 | 0.093 | 0.114 | **0.490** |
| Transport interchanges | 0.272 | 0.035 | 0.277 | 0.432 | −0.196 | **0.485** |
| Explained variance (in percent): for factor | 27.783 | 8.885 | 8.363 | 7.018 | 6.020 | 5.658 |
| Cumulated | 27.783 | 36.668 | 45.031 | 52.049 | 58.069 | 63.727 |
| Cronbach's alpha | 0.815 | 0.727 | 0.801 | 0.606 | 0.605 | 0.714 |

Factor extraction method—Main components. Rotation method—Equamax with Kaiser normalisation. KMO = 0.808; i Bartlett's test of sphericity: $\chi^2$ (171) = 1377.3. Source: own elaboration.

The next subscale (F4—active mobility infrastructure for cycling and walking) covers three areas: cycle routes, pedestrian routes, and woonerfs, while the fifth (F5—intra-city infrastructure for individual car transport) considers four aspects: car parks in the city and on the outskirts, roads in the city, and charging infrastructure for electric vehicles. The last group (F6—solutions for intermodal intra-urban travel) is made up of five solutions: interchanging between public transport vehicles and car rental by the minute, public transport stops, tram tracks, and transfer stations. Mobile transport apps are also included in this group, although this issue is also quite strongly related to the first factor (however, for F6 the factor load is higher: 0.490). This is due to the need to install transport apps

in order to use modern mobility services such as ridesharing vehicles. The reliability of the last subscale is good (Cronbach's alpha coefficient of 0.714); for the other two (F4 and F5) it is satisfactory, and the degree of explanation of the variance of the latent variable exceeds 5%. In total, the six extracted factors explain nearly 64% of the variation in the latent variable (against the required minimum of 50%). In conclusion, the Cronbach's alpha coefficient and the exploratory factor analysis confirm that the proposed tool can be used to measure the assessment of the development of the city's transport system.

4.2.2. Assessment of the Benefits of the Development of Intelligent Transport Systems by the Residents of Lodz

The benefits of developing intelligent transport systems were assessed for five aspects (Table 8). All of them are rated highly on a scale from 1 to 7, with a median reaching 5–6 and a mean of 4.44–5.43. The greatest benefits are perceived in relation to improved road safety (M = 5.43, SD = 1.31), with the lowest percentage of lowest responses here (1–2) at only 4% compared to over 10% for reduced pollution and congestion. Improved travel comfort (Me = 6, M = 5.38) and reduced travel time (Me = 6, M = 5.34) were also highly rated.

**Table 8.** Benefits of the intelligent transport systems.

| No. | Percentage of Responses for the Answer | | | | | | | | Statistics | | | rho | | | | |
|-----|------|-----|------|------|------|------|------|------|------|------|------|---------|---------|---------|---------|----|
|     | 1    | 2   | 3    | 4    | 5    | 6    | 7    | n.d. | M    | Me   | SD   | X1      | X2      | X3      | X4      | X5 |
| X1  | 2.4  | 4.8 | 5.6  | 3.6  | 32.4 | 29.6 | 21.6 | 0.0  | 5.34 | 6.00 | 1.46 | 1       |         |         |         |    |
| X2  | 0.0  | 4.0 | 5.6  | 8.0  | 32.4 | 25.6 | 24.0 | 0.4  | 5.43 | 5.00 | 1.31 | 0.399 * | 1       |         |         |    |
| X3  | 1.2  | 9.2 | 6.8  | 6.0  | 29.6 | 23.6 | 23.6 | 0.0  | 5.19 | 5.00 | 1.58 | 0.515 * | 0.448 * | 1       |         |    |
| X4  | 1.2  | 5.6 | 6.0  | 5.2  | 30.0 | 27.6 | 24.4 | 0.0  | 5.38 | 6.00 | 1.45 | 0.502 * | 0.456 * | 0.609 * | 1       |    |
| X5  | 3.6  | 9.6 | 18.8 | 15.6 | 20.0 | 20.4 | 10.4 | 1.6  | 4.44 | 5.00 | 1.65 | 0.361 * | 0.424 * | 0.458 * | 0.406 * | 1  |

M—mean, Me—median, SD—standard deviation, * $p < 0.05$. Source: own elaboration

Benefits of the development of intelligent transport systems (X1–X5) are high (X3 vs. X4, X1 vs. X3 and X4) or moderate. Correlations between all items of "benefits" scale are statistically significant and positive (Table 8).

4.2.3. Summary Assessment of the Development of a Sustainable and Intelligent Transport System and Benefits of the Development of ITS

The above discussed 19 elements (Y1–Y19) help in carrying out a reliable measurement (Cronbach's alpha coefficient reaching 0.847) to arrive at the overall assessment of the development of a sustainable and intelligent transportation system in the city. As each of the partial variables could take values from 1 to 5, the total assessment of the development of the transport system (variable "Assessment") can assume values from 5 to 95. Additionally, speaking about benefits, the five elements adopted for the survey allow for their reliable assessment (Cronbach's alpha coefficient of 0.801). The summary assessment of benefits was determined as the sum of the scores obtained for variables X1–X5. As each of the partial variables could take values from 1 to 7, the summary benefit score (variable "Benefits") could take values from 7 to 35. The distributions of both variables are presented in Table 9.

**Table 9.** Descriptive statistics for the summary assessment of the development of the transport system and benefits of developing an ITS.

|            | n   | Min | Max | M     | Me    | SD   | S     |
|------------|-----|-----|-----|-------|-------|------|-------|
| Assessment | 250 | 19  | 76  | 49.14 | 50.00 | 8.21 | −0.50 |
| Benefits   | 245 | 9   | 35  | 25.80 | 26.00 | 5.55 | −0.60 |

M—mean, Me—median, SD—standard deviation, S—skewness. Source: own elaboration.

Benefits were highly rated, with max = 35, and the mean and median reaching 26, with relatively low variation in scores (SD = 5.55) and low skewness of the distribution

(S = −0.60). The rating for the development of the transport system is lower; no one selected the highest possible rating (max rating is 76 against 95). Half of the people gave a rating no higher than 50, the average being 49 (also with relatively low score variation and low skewness of the distribution).

A comparison of the two above-mentioned variables by metric characteristics shows that there are no significant differences across all the characteristics, as gender, age, income, and car ownership do not significantly differentiate the assessment of transport system development and benefits. What is significant, however, is the moderately strong correlation between benefits and the assessment of the development of the transport system, where Pearson's linear correlation coefficient is r = 0.380 ($p < 0.001$). Significantly higher scores for the development of the transport system are given by respondents who see greater benefits in the implementation of intelligent transport systems, but the assessment of benefits is also higher for residents who rate the development of the transport system in the city higher.

The estimates of the linear regression model confirm that the assessment of the development of a sustainable and intelligent transport system is significantly higher for people who rate the benefits of intelligent transport systems higher (model 1, Table 10). The inclusion of control variables in the model (gender, labour force participation, and age) only slightly increases the degree of explanation of the variation in scores (adjusted $R^2$ = 0.146), which is not surprising given that, at the significance level α = 0.05, none of the "metric" variables are significantly related to the evaluation of the development of the city's transport system (model 2, Table 10). Ceteris paribus (i.e., assuming "metric" characteristics for all respondents are the same), the relevance of benefits remains statistically significant ($p < 0.001$); the assessment of the development of the city transport system is significantly higher for people who rate the benefits higher. The standardised Beta coefficient confirms that the relevance of benefits is significant in this regard and clearly higher than for the other variables (Beta = 0.375). Age also plays a role; at a significance level of 0.10, those aged 31–40 and over 40 rate the development of the city transport system significantly higher than those under 20.

**Table 10.** Results of linear regression analysis: *y*–assessment of smart transport solutions.

| | B | S(B) | Beta | t | P | VIF | ANOVA | $R^2_{sk}$ |
|---|---|---|---|---|---|---|---|---|
| **Model 1** | | | | | | | | |
| Constant | 34.687 | 2.319 | | 14.961 | <0.001 * | | F(1; 243) = 40.916; | 0.141 |
| Benefits | 0.562 | 0.088 | 0.380 | 6397 | <0.001 * | 1.000 | *p* < 0.001 * | |
| **Model 2** | | | | | | | | |
| Constant | 34.296 | 2.823 | | 12.150 | <0.001 * | | | |
| Benefits | 0.556 | 0.090 | 0.375 | 6.192 | <0.001 * | 1.020 | | |
| Gender [1] | −0.826 | 1.046 | −0.050 | −0.789 | 0.431 | 1.110 | | |
| Labour force participation [2] | −0.682 | 1.294 | −0.040 | −0.527 | 0.599 | 1.619 | F(7; 231) = 6.831; | 0.146 |
| Car ownership [3] | −1.620 | 1.154 | −0.095 | −1.404 | 0.162 | 1.271 | *p* < 0.001 * | |
| Age [4] | | | | | | | | |
| 21–30 | 1.882 | 1.925 | 0.101 | 0.978 | 0.329 | 2.955 | | |
| 31–40 | 5.105 | 2.699 | 0.171 | 1.891 | 0.060 [t] | 2.273 | | |
| 40+ | 4.764 | 2.639 | 0.176 | 1.805 | 0.072 [t] | 2.653 | | |

Reference group: [1] men, [2] inactive, [3] do not own a car, [4] under 20. B—regression coefficient, S(B)—standard error for B, Beta—standardised regression coefficient, t—t-Student statistics, p—probability in *t* test, $R^2_{sk}$—standardised coefficient of determination; * $p < 0.05$, [t] $p < 0.10$. Source: own elaboration.

The statistical properties of the constructed model are correct (ANOVA shows that the coefficient of determination in the population is significantly different from zero, VIF < 10 confirms the lack of collinearity of the explanatory variables, and the analysis of the distribution of the residuals confirms that the assumption of normal distribution of the random component has been fulfilled). At the same time, the value of the coefficient of

determination suggests that the score is largely determined by factors other than those included in the model (approximately 85% of its variability is due to other factors).

## 5. Discussion

In this paper we present and test two methods for analysing a smart and sustainable urban mobility transition. The first one concerns public policy, and consists of a content analysis of public policies that address the development of sustainable urban transport. It is used to find out whether the programmes address urban mobility transition in a comprehensive and integrated way, while assuming its smart and sustainable development andintegrating technological, environmental, spatial, social, economic, and planning aspects and solutions. In addition, it examines whether the strategies provide mechanisms to ensure that the results are effectively achieved in a planned manner through methods that influence the attitudes of city dwellers and increase their approval of sustainable and intelligent transport solutions.

The analysis of the national, regional, and local policies for smart urban mobility transition in Poland carried out in this paper, in accordance with the above-mentioned assumptions, allowed us to draw conclusions about the comprehensiveness of these policies at the national level. National transport strategy sets out a stable, mature framework for urban mobility management, well operationalised at the level of objectives and measures. Furthermore, the strategy integrates urban transport issues with transport sustainability and digitalisation. On the other hand, it does not explicitly assume the participation of citizens in the development of urban transport policy, but, by recommending that Polish cities develop SUMPs, only indirectly indicates the need for it. Meanwhile, an important prerequisite for effective transport policy is social acceptance [57], which requires well-developed mechanisms designed to involve residents in the planning and implementation of activities [58]. It is worth mentioning that, at the national level, EU-funded measures are being taken to support cities in the preparation and implementation of these plans. These result from Poland's participation between 2017 and 2019 in the PROSPERITY (Prosperity through innovation and promotion of Sustainable Urban Mobility Plan) project funded by the Horizon 2020 Programme, which aimed to promote and develop SUMPs [59]. On the other hand, at the regional (development strategy for the Lodzkie Voivodeship) and local levels, with the latter being examined in the example of the city of Lodz, we noted a failure to include instruments that would influence transport decisions of the inhabitants, and the involvement of the inhabitants in the shaping of transport policy in development strategies. Neither do these strategies provide for the development of a SUMP. Therefore, our analysis highlights the lack of full vertical coordination of the Lodz transport policy with regard to an intelligent and sustainable urban mobility transition.

A prerequisite for the transition to sustainable and smart mobility rests on residents' understanding its benefits. Studies show that a positive attitude towards sustainable modes of travel and seeing them as suitable for daily journeys influence people's decisions about using them [60]. At the same time, there is a discrepancy between an individual's willingness to switch to green transport and the actual change in travel behaviour when there are several external constraints, such as poor availability of public transport services or their poor quality, and a lack of information on how to save resources and reduce emissions by using green transports. Providing information through targeted information campaigns on the benefits of sustainable transport behaviour, including personal benefits such as saving money and time spent on finding a parking spot, is seen as a tool to motivate individuals to translate their behavioural will into action [61,62].

Communicating the environmental and social challenges of modern urban transport and the potential of sustainable and intelligent mobility in dealing with these challenges should be one of the key impacts of public policy. Unfortunately, the analysed strategies focus on equally important aspects of infrastructure investments, improvement of public transport fleets, or implementation of smart solutions, while neglecting the information policy measures indispensable for developing sustainable transport behaviour of inhabi-

tants. The lack of a comprehensive approach and integration of the solutions planned in the strategies also manifested in the omission of pricing and fiscal tools, which, according to researchers, also determine the behaviour of transport system users [63].

The second method used to investigate the smart and sustainable urban mobility transition discussed in this article consists of studying how city residents perceive such a transition. For this purpose, a scale was proposed to measure the development of a sustainable and intelligent transport system for the city, which takes into account 19 items, grouped on the basis of exploratory factor analysis into six categories: (1) intelligent transport systems and services, (2) public transport passenger information system, (3) road infrastructure of national importance in the agglomeration surroundings, (4) active mobility infrastructure for cycling and walking, (5) intra-city infrastructure for private car transport, including electric vehicles, and (6) solutions for intermodal intra-city travel. This tool takes into account the smart and sustainable transport solutions discussed in the introductory section and resulting from the literature review.

The results of the analysis of the local policy in the field of urban transport in Lodz, the characteristics of implemented smart transport solutions in the city, and survey results show that the transition of the Lodz transport system towards an intelligent and sustainable one is taking place in a non-integrated and moderate manner. Although emphasis is placed on putting modern digital solutions (ITS, shared mobility systems, and transport applications) in place, shortcomings in the expansion and insufficient modernisation of the traditional infrastructure for public and individual transport, as well as pedestrian and bicycle infrastructure persist and are accompanied by insufficient integration of modern and traditional transport services. This is confirmed by respondents' average and low ratings for these aspects.

Of note, according to the TomTom Traffic Index 2021, Lodz is the most congested city in Poland and one of the most congested in Europe [64]. Despite the implementation of ITS and shared mobility systems, congestion remains high, meaning active forms of urban mobility (walking and cycling) and public transport remain unattractive to inhabitants, and public transport services are insufficiently integrated with other urban mobility options. The poor integration of urban public transport with parking and shared mobility systems is confirmed by respondents' assessments. Similar conclusions on insufficient integration of shared mobility systems with the public transport in Lodz have been formulated in studies on the cycle share scheme operated by the city [65,66]. A review of studies conducted by Huang and Loo [67] shows that, despite their high potential, the effectiveness of technological solutions in alleviating congestion is still uncertain. Recent analyses of congestion in US cities again prove that an effective congestion mitigation strategy must include the availability of alternative modalities, including efficient public transport [68]. Cheng et al. [69], on the other hand, provide evidence that ITS have a significant impact on congestion reduction, and the effect depends on road supply and public transit services. On the other hand, analyses by Okafor et al. [70] point to cycling and walking infrastructure as a critical factor in the development of smart mobility. The research results cited above point to the need for integrated actions, involving the implementation of digital transport solutions and the development of alternative transport options to car-based individual transport, such as public urban transport and infrastructure for active transport.

Furthermore, a sustainable mobility policy should include "pull measures" that increase the attractiveness of public transport, e.g., through technological innovations or its appropriate promotion [71]. In the case of Lodz, public transport promotion and education activities on smart, sustainable mobility targeting the inhabitants are incidental and not provided for in the city's development strategy. The Sustainable Development Plan for Public Transport in Lodz only signals the need to inform residents about the environmental benefits of switching to public transport. According to our findings, such measures are indispensable as respondents' perceptions of environmental benefits are slightly lower than those of improving travel experience and safety through the implementation of innovative transport

solutions. Results of other studies [72] suggest that raising residents' environmental awareness of smart mobility should lead to a greater willingness to be environmentally friendly.

In this article, we have confirmed the existence of a significant relationship between the perception of the benefits of the development of intelligent transport systems and the assessment of the level of development of a sustainable, intelligent transport system for a city. This relationship is positive. Respondents who rate the above benefits higher, rate the sustainability and intelligence of the transport system higher, and vice versa. In light of these findings, we can conclude that growing awareness of the benefits of a transition towards sustainable and smart mobility coincides with progressing advancement of the transition. Increased awareness of the benefits may further determine the use of digital and smart solutions. Such a link between perceived usefulness and an attitude towards the adoption of smart mobility systems has already been confirmed by other researchers [73]. We postulate further research in this area. Our analyses do not confirm the relevance of variables such as respondents' age, gender, income, labour market participation, or car ownership for the relationship between perceived benefits and the development of the city's transport system. We also believe that looking for other determinants is a valid direction for future research. Due to the limitation of our study, i.e., a relatively small research sample (250 respondents, the margin of error—6%), we suggest the validation of our research approach on a bigger group of respondents. There is also another limitation to our study, as we analysed only two dimensions (political and social) of the environment in which an urban transport system is encapsulated, leaving other relevant perspectives (e.g., financial, legal, urban planning, spatial, or technical) aside.

A weakness of the policy dealing with smart urban mobility transition in Lodz lies in the failure to apply a participatory model, widely disseminated in the EU in the SUMP formula, in its formulation. Lodz authorities have not planned to develop such a plan in the city's development strategy; nor are there any other forms of resident involvement in the shaping of local transport policy (with the exception of opinion polls on the quality of public transport services), in the co-rendering of public services, or in the co-designing of urban mobility services. This is a major shortcoming, as research findings show that the success of smart mobility projects in achieving not only instrumental but also social value depends on placing the user at the centre of the design process [74]. Digital mobility solutions should be tailored to user needs and, in the context of varying digital competences, also to their capabilities. In the case of Lodz, this is particularly relevant given the dynamic ageing of the city's population and the digital exclusion among older adults highlighted in literature and manifested in the limited use of the Internet and smartphones [75–77]. In addition, mobility policy is characterised by a high degree of complexity and participatory processes should allow for conflicting interests to be taken into account [71].

## 6. Conclusions

Our study leads to several theoretical, empirical, and practical implications. Firstly, it bridges the knowledge gap with regard to the policy pursued by the local and central government in Poland vis-à-vis the transition towards sustainable and smart mobility. Secondly, we examine public policy assumptions by identifying provisions included in strategies drafted at central government, regional, and local levels which so far have not been the subject of scientific research on sustainable mobility. The tool we have developed for this purpose can be applied to conduct similar research in other countries. Additionally, the tool tested in this paper that is used to assess the development of a sustainable and intelligent transport system for a city can be used by other researchers. The general conclusion from our studies is that the measures programmed within public policy to promote sustainable mobility are quite fragmented. Thus, our recommendation for politicians, especially local and regional ones, is to focus on coordinating activities so that they integrate infrastructural, technological, organisational, natural, social, and behavioural aspects. In view of the lack of a user-orientated urban transport system that we have identified, a well thought out information and education policy is particularly

important. It should, above all, focus on environmental risks and their consequences for the health and well-being of individuals, the advantages of sustainable mobility, or individual gains from choosing green modes of transport for daily journeys. The information should also be useful to users to improve their daily mobility. This includes the promotion of public transport services and reliable and timely information on fares, ticketing options, and timetables, and the integration of different modes of transport. In addition, improving the quality of transport services in the city requires a systematic survey of users' opinions on the quality of the services offered and a study of their ability to use transport solutions. By this we could make sure that the expected solutions are designed and implemented while residents are properly informed and educated on how to use them.

**Supplementary Materials:** The following supporting information can be downloaded at: https://www.mdpi.com/article/10.3390/en16010143/s1, Table S1: Correlation between items of the assessment of the development of sustainable and intelligent transport system in Lodz.

**Author Contributions:** Conceptualization, J.P. and A.P.-K.; methodology, J.P. and A.P.-K.; software, J.P. and A.P.-K.; validation, J.P. and A.P.-K.; formal analysis, J.P. and A.P.-K.; investigation, J.P. and A.P.-K.; resources, J.P. and A.P.-K.; data curation, J.P. and A.P.-K.; writing—original draft preparation, J.P. and A.P.-K.; writing—review and editing, J.P. and A.P.-K.; visualization, J.P. and A.P.-K.; supervision, J.P. and A.P.-K.; project administration, J.P. and A.P.-K. All authors have read and agreed to the published version of the manuscript.

**Funding:** This research received no external funding.

**Data Availability Statement:** Not applicable.

**Acknowledgments:** We remain grateful to Justyna Wiktorowicz from the University of Lodz for her support in statistical methodology.

**Conflicts of Interest:** The authors declare no conflict of interest.

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
