# Peer review of "Public Policy and Citizens’ Attitudes towards Intelligent and Sustainable Transportation Solutions in the City—The Example of Lodz, Poland"

_energies, doi:10.3390/en16010143_

Round 1

Reviewer 2 Report

The topic is important but the manuscript should be improved:

1. English writing should be improved significantly.

2. The quality of literature review is poor. The term "intelligent and sustainable transportation solutions" is very vague and unclear.

3. statistical data is not well structured. Only 250 people have been interviewed. There is a need to carry out at a larger scale (e.g. 2,000 responders).

4. The features to define intelligent and sustainable transportation solutions are unclear. This must be revised significantly.

5. Data features should be better analysed. The data flow and interconnectedness are not analysed at all.

6. There is no clear link about the content in this manuscript to Energies Journal.

7. Life cycle aspects are missing completely.

8. Sustainability pillars are missing completely.

Round 2

Reviewer 1 Report

The article has been corrected - I approve its publication.

Reviewer 2 Report

Ok to accept.